# Wood formation of drought-resistant *Eucalyptus cladocalyx* under cyclical drought treatment

Gugu Gama , Kim C. Martin and David M. Drew

Department of Forest and Wood Science, Stellenbosch University, Stellenbosch, South Africa

## Original Research Article

cyclic drought; wood formation; xylem response.

**Corresponding author:**
David M. Drew;
Email: drew@sun.ac.za

**Associate Editor:**
Félix Hartmann

## Abstract

*Eucalyptus cladocalyx*, known for its drought tolerance, has complex wood anatomy influenced by environmental conditions. This study investigated the xylem response of *E. cladocalyx* seedlings to cyclic drought stress compared to continuous irrigation. Seedlings were subjected to alternating drought and watering cycles, and their growth, xylem traits and cambial activity were monitored. Continuously irrigated seedlings exhibited greater height and stem diameter growth than periodically irrigated ones. Xylem response between the periodic and continuous irrigations showed no significant differences. Vessel and fibre features showed significant temporal variation, with substantial interaction between treatment and time for vessel area, fibre area and fibre thickness and not for vessel frequency. The cambium remained active under drought conditions, indicating resilience. Overall, anatomical properties varied complexly and inconsistently across drought cycles, likely due to differences in drought intensity, strategies and genetic factors.

## 1. Introduction

The wood of *Eucalyptus* has a complex anatomy, which influences its commercial and industrial value. The wood consists of numerous small fibres (Carrillo et al., 2018), solitary vessels, vasicentric tracheids, fibre tracheids and has a wide wood density range (Carrillo et al., 2018; José Barotto et al., 2017). These properties are outcomes of xylem development and xylogenesis dynamics and are often affected by changes in environmental conditions (Paux et al., 2005). Because of its responsiveness to changing environmental conditions, *Eucalyptus* wood anatomy is complex when studied at the tissue and cellular levels (Drew et al., 2010; Franks et al., 1995), reflecting changes in xylem development and other growth-related processes. These changes can be seen in features such as cell size, shape, wall structure, texture, cell type and arrangement, as well as in the chemical composition of secondary cell wall fibres (Carrillo et al., 2018; Paux et al., 2005). Changes in xylem development inevitably affect the critical final xylem tissue traits (Li et al., 2000) and thus the quality and quantity of the resulting wood.

The capacity of the xylem to continue conducting water can reduce under drought conditions, as embolism increases, which in turn reduces productivity and likelihood of tree survival (Van der Willigen & Pammenter, 1998; Franks et al., 1995). On the other hand, xylem can also prevent drought vulnerability (Van der Willigen & Pammenter, 1998) through complex changes in vessel diameter (wider or smaller vessels) as well as changes in pit membrane thickness (Lens et al., 2022), thereby, limiting hydraulic failure and leading to greater drought resistance (Bouda et al., 2022). Although extensive research has been conducted on the effects of drought, little is known about how changing environmental conditions affect xylogenesis and the underlying developmental dynamics (Farooq et al., 2023; Valenzuela et al., 2021; Li & Jansen, 2017). Even less is known about these dynamics in drought-tolerant eucalypts (Valenzuela et al., 2021) that are often difficult to study because of the lack of standardised methods (Chiang & Greb, 2019; Liu et al., 2018; Huang et al., 2014; Schmitt et al., 2013). Some *Eucalyptus* species can be productive and of great value even when grown in semi-arid conditions, such as *Eucalyptus cladocalyx* (Du Toit et al., 2017). A potentially commercially important species such as *E. cladocalyx*, which has adapted to dry conditions and developed responsive mechanisms to withstand both short- and long-term droughts (Mora et al., 2009; Akhter et al., 2005) is, however, understudied

(Li et al., 2000; Myers & Landsberg, 1989). Furthermore, no work has been done on the xylem responses and resulting wood properties of the valuable *E. cladocalyx* under varying drought conditions (Valenzuela et al., 2021; Li et al., 2000).

*Eucalyptus cladocalyx* (Sugar gum) thrives in semi-arid, sub-humid and xeric environments with low annual precipitation (< 800 mm) and regular droughts (Héroult et al., 2013). This species is also intricately linked to its genetic makeup and possesses a significant portion of differentially expressed genes under water scarcity conditions that are related to cellular metabolism, including MFS genes (Valenzuela et al., 2021). These processes play a vital role in *E. cladocalyx's* ability to withstand harsh environmental conditions and grow in dry lands (Mora et al., 2009; Marcar et al., 2002; Rawat & Banerjee, 1998). Even when planted outside its natural habitat, it retains good form and wood characteristics while also being fast-growing (Carrillo et al., 2018; Du Toit et al., 2017; De Lange et al., 2013). Under arid conditions, *E. cladocalyx* produces high-quality hard and structural timber (Valenzuela et al., 2019; Carrillo et al., 2018; De Lange et al., 2013; Merchant et al., 2006), which is ascribed to its high-density wood (Carrillo et al., 2018), the rigidity of its microfibrils (Valenzuela et al., 2019) and tendency to develop a straight form without branched stems (Valenzuela et al., 2019). *E. cladocalyx* is also considered resistant to embolism, which is linked to its drought tolerance (Carrillo et al., 2018). Yet little is known about the xylem physiology of *E. cladocalyx* and its response to drought.

In summary, *E. cladocalyx* possess useful physical and mechanical properties that are heavily influenced by vessels and parenchyma, as well as fibre width, cell wall thickness and chemical composition (Valenzuela et al., 2019; Carrillo et al., 2018). Knowledge of such elements will give a deeper understanding of the processes of xylogenesis for the improvement of wood properties (Drew & Pammenter, 2007; Ridoutt & Sands, 1993). It is generally observed that vessel and fibre features in *Eucalyptus* wood show a strong negative correlation with water deficit, showing changes in vessel frequency and thicker cell walls (Barbosa et al., 2019). Thus, understanding the xylem anatomy of *E. cladocalyx* is important for elucidating adaptive mechanisms under a variety of environmental conditions to link drought tolerance to anatomical variation (Sorce et al., 2013; Rossi et al., 2011; Plomion et al., 2001). Considering this and the economic importance of *E. cladocalyx*, we investigated how xylem in *E. cladocalyx* seedlings responded to multiple short drought stress cycles. To clarify the relationship between xylem developmental traits and periodic drought cycles, growth responses and variability of xylem anatomy and dynamics were assessed in seedlings exposed to cyclic versus consistent watering regimes. The following three major research questions were addressed:

1. Were there observable differences in xylem properties between trees exposed to cyclical short-term drought and those that were not?
2. What changes in xylem properties occurred during and after drought (cell size, shape, vessel diameter and fibre area)?
3. How easily could the final xylem properties be linked to the cambial zone dynamics during the course of the experiment?

## 2. Materials and methods

### 2.1. Site and growth conditions

The study was conducted in an experimental greenhouse at the Department of Forestry and Wood Science, Stellenbosch University, South Africa (33°55′35.7″S 18°51′59.5″E). The experiment comprised of 240 *E. cladocalyx* seedlings. The seeds were obtained from the so-called 'tree A1', a selected superior tree, in the Coetzenburg seed orchard in Stellenbosch. The mother tree comes from the Kerksbrook SPA provenance and seed lot number 20595 (Botman, 2010). After germination, individual seedlings were transferred to 1.5-L planter bags containing 50% composted pine bark and 50% river sand and placed in a shaded area (50% net coverage). All seedlings received fertilizer as well as foliar fertilizer prior to the experimental treatment. Although pre-dawn leaf water potential (LWP) measurements were the primary metric by which drought level was regularly assessed, periodic soil water content measurements were made with a basic soil probe, for reference. In the trees that were periodically droughted, volumetric soil water content (VWC) ranged between 1% and 10%, with an average of 5.4% with standard deviation of 3.5%. For the control trees, which were deliberately not exposed to limiting water conditions, the VWC ranged between 7% and 10%, with an average of 8.8% with standard deviation of 1.0%.

### 2.2. Experimental design and repeated drought treatment

The seedlings were moved to the greenhouse when they reached an average height of 50 cm. Two irrigation treatments (continuous irrigation (CI)) vs. periodic irrigation (PI))) were applied to 60 seedlings per treatment for 10 weeks (22 March to 1 June 2017) to assess the response to cyclic drought stress. Seedlings in both treatments were arranged in a randomised block design on the east and west sides of the rain-free greenhouse (Figure 1). Each block had 60 seedlings where blocks B and C represented the CI treatment and blocks A, and D represented the PI treatment.

To achieve the CI treatment, seedlings were continuously watered to field capacity daily. The PI treatment was designed in such a manner to subject seedlings to a drought treatment (DT), followed by a watering treatment (WT) on a cyclic basis for a total of six cycles. These cycles were numbered from 1 to 6, as summarised in Table 1.

To achieve the WT initial state, all trees (including the PI trees) were watered daily to field capacity for 2 weeks prior to the first drought event (22 March 2017). Thereafter, DT was achieved by withholding water for periods of 3–7) (Table 1), but this was adjusted based on visible signs of stress, thereby ensuring that seedlings experienced acute drought stress, but did not perish.

### 2.3. Measurements

An Easylog USB data logger (Lascar Electronics Ltd, PA, US) was suspended 2 m above the ground in the middle of the experimental setup to record temperature and relative humidity every 30 min throughout the trial. From the start of the first drought treatment (22 March 2017), 12 trees in each block (a total of 48 seedlings for both CI and PI) were earmarked for continuous diameter and height measurements and were not sampled until the end of the experiment. During the experiment, as well as on experiment termination, the diameter of each stem (n = 12 per block) was measured daily at a position 2 cm above soil using a digital Vernier calliper with a 0.01 mm resolution (Mitutoyo Corporation, Japan). In addition, the height of the 12 seedlings in each block was measured daily with a tape measure from the base to the tip of each seedling.

To determine the pre-dawn LWP ($\Psi_{PD}$), a Scholander-type pressure chamber (Skye Instruments Limited, Wales) was used to

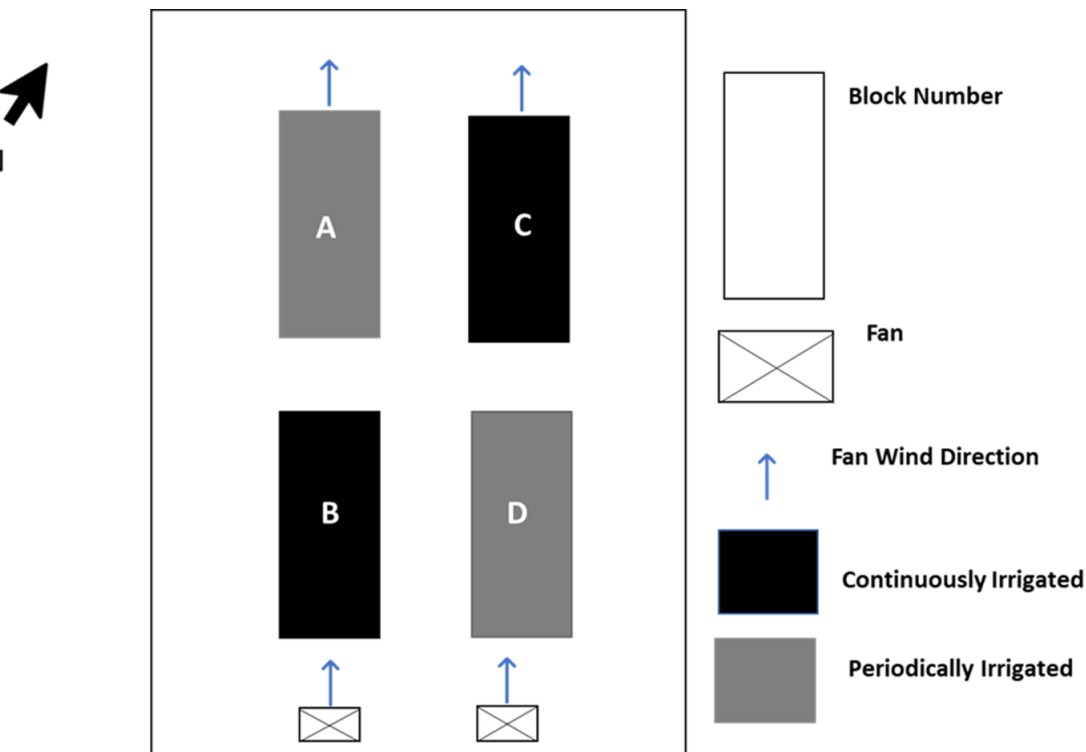

**Figure 1.** Layout of trial in the greenhouse. Each block contained 60 seedlings at the start of the experiment, of which 12 trees per block were retained for continuous measurements until the end of the experiment. Throughout the experiment, destructive sampling of seedlings was carried out with two seedlings per block sampled at each timepoint.

**Table 1.** The cycles of drought applied to the trees, with the codes for the different drought/watering cycles

| Cycle | Treatment | Start date | Number of days | Days cumulative at start of each cycle |
|---|---|---|---|---|
| DT1 | Not watered | 8 April | 5 | 19 |
| WT1 | Watered | 13 April | 3 | 24 |
| DT2 | Not watered | 16 April | 3 | 27 |
| WT2 | Watered | 19 April | 3 | 30 |
| DT3 | Not watered | 22 April | 6 | 33 |
| WT3 | Watered | 28 April | 3 | 39 |
| DT4 | Not watered | 5 May | 7 | 42 |
| WT4 | Watered | 8 May | 5 | 49 |
| DT5 | Not watered | 13 May | 6 | 54 |
| WT5 | Watered | 19 May | 4 | 60 |
| DT6 | Not watered | 23 May | 6 | 64 |
| WT6 | Watered | 29 May | 4 | 70 |

measure the water potential of two to three healthy, fully expanded leaves using the method described by Zuecco et al. (2022). The measurements were taken before sunrise. In brief, leaves were cut cleanly where the petiole meets the stem and immediately placed into the pressure chamber. A nitrogen gas cylinder was then used to gradually apply pressure (measured in MPa) to each leaf to the point where the sap started to exude from the severed stem, and this pressure was taken as $\Psi_{PD}$.

## 2.4. Sampling

Every third day, two trees from each block (excluding the 12 regularly measured trees) were randomly selected for destructive sampling to determine whether the treatments influenced pre-dawn LWP, biomass and the cambial/xylem developmental responses (Figure 2). Samples were consistently on each occasion before water application. The samples were fixed in Paraformaldehyde fixative (4% formaldehyde in water) and kept in a refrigerator at 4°C until sectioning. To determine whether the treatments influenced the wood anatomy and xylem functional traits of *E. cladocalyx*, 12 seedlings from each block were finally collected at the end of the experiment. A 2 cm length of each stem (starting from the bottom) was removed from each seedling and stored in formaldehyde alcohol acetic acid (FAA) solution (Merck, Darmstadt, Hesse, Germany; F8775, 1070172511, 1018302500) for microscopic analysis.

## 2.5. Preparation of thin sections for microscopy

**2.5.1. Samples collected during the experiment.** Samples that were collected during the experiment, that is, those collected every third day, were further processed into smaller sub-samples (one-eighth of the circular surface) of each stem. Each sample was subsequently rinsed five times for 3 min each in 0.1 M phosphate buffer solution. These samples were then dehydrated in a series of ethanol concentrations (30% (v/v), 50% (v/v), 70% (v/v), 96% (v/v) and 100% (v/v)) as described by Glauert and Lewis (1999). Thereafter, the samples were embedded in an upright orientation in Durcupan resin (DurcupanTM ACM components A, B, C and D, M epoxy resin) in a capsule mould. The samples were first

**Figure 2.** Sampling of seedlings throughout the experimental trial. Each block contained 12 seedlings that were continuously measured throughout the experiment (only sampled at the end of experiment). Every third day, two seedlings were randomly selected for destructive sampling. Water potential was measured from the randomly selected seedlings before sampling.

**Table 2.** Average temperatures and relative humidity measured in the experimental greenhouse during each cyclic event, where the trees (n = 240) were droughted until signs of visible stress (drought treatment (DT)), followed by watering them up to field capacity daily (watering treatment, WT) for six cycles

| Cycle | Min T(°C) | Max T(°C) | Avg T T(°C) | Min RH (%) | Max RH (%) | Avg RH (%) |
|---|---|---|---|---|---|---|
| DT1 | 14 | 45 | 25 | 22 | 85 | 58 |
| WT1 | 10 | 28 | 18 | 35 | 84 | 70 |
| DT2 | 12 | 43 | 23 | 19 | 76 | 49 |
| WT2 | 14 | 43 | 24 | 19 | 71 | 49 |
| DT3 | 12 | 42 | 23 | 16 | 86 | 55 |
| WT3 | 15 | 44 | 30 | 19 | 79 | 38 |
| DT4 | 12 | 38 | 22 | 31 | 87 | 63 |
| WT4 | 12 | 33 | 19 | 30 | 81 | 60 |
| DT5 | 8.5 | 39 | 20 | 16 | 83 | 54 |
| WT5 | 10 | 36 | 18 | 29 | 82 | 65 |
| DT6 | 9.5 | 36 | 18 | 22 | 90 | 70 |
| WT6 | 10 | 35 | 18 | 20 | 79 | 55 |

Min T – minimum temperature measured during a cycle period, Max T – maximum temperature measured during a cycle period, Avg T – average temperature measured during a cycle period Min RH – minimum relative humidity measured during a cycle period, Max RH – maximum relative humidity measured during a cycle period, Avg RH – average relative humidity measured during a cycle period.

trimmed to remove excess resin using a minora blade (Gillette, Russia) and hereafter sectioned to 1 μm in a transverse orientation using a Leica UC7 ultra-microtome (Leica Microsystems, Wetzlar, Germany). The sections were stained with toluidine blue (0.05% in distilled water) for 1 min and rinsed with water for 2 min. Finally, the sections were mounted on glass slides for microscopy using uniLAB® DPX mountant (Merck). A Leica light microscope (Leica Microsystems) was used to obtain bright-field images with 20× objective. Images were also obtained for at least three portions of the cambial zone, with the same position with/without polarisation.

**2.5.2. Samples collected at the end of the experiment.** To investigate the effects of the treatments on the xylem functional traits of *E. cladocalyx* over the 75-day experiment, a smaller subsample (one-eighth of the circular surface) was cut from each of the 2 cm stem sections. Each sample was dehydrated, embedded, softened, sectioned and stained using the method described by Keret et al. (2024). In brief, the samples were stored in FAA, infiltrated using HistoCore Pearl Tissue Processor (Leica Microsystems) and embedded using paraffin wax (Merck; 411663). Hereafter, the paraffin blocks were trimmed (at 8um) and sectioned using a rotary microtome (Leica Biosystems, Deer Park, Texas, USA; RM-2245) at a thickness of 5–6 μm. The transverse sections were then placed onto microscope slides, dried and stained with Safranin–Alcian blue (Merck; 84120 and A5268). Full images from bark to pith (width=1825 pixels and height=7803 pixels) were obtained using an Eclipse E400 Microscope (Nikon Instruments, Melville, New York, USA) fitted with a 20× objective and 5-megapixel Nikon DS-Fi2 camera. The images were ensured to be of clear and good quality and same length and width.

### 2.6. Software and image processing

Qupath (Bankhead et al., 2017) and R (R Core Team, 2023) were used to estimate metrics for subsequent analysis (see supplementary data). Qupath was used to detect and classify the xylem cells as described in Keret et al. (2024). In brief, a project was created in Qupath, and the image dataset was loaded and set to pixel size based on the camera and microscope settings, followed by defining regions of interest (ROIs) in the image. Cells within the ROI were detected using adjustable parameters, and a machine learning classifier (Random Trees classification) was trained to categorise cells into predefined classes like fibres and vessels. Hereafter, the detection results of anatomical properties were viewed and exported as a csv document. Detection results included parameters such as cell area, nucleus area and centroid positions of cells for fibres and vessels.

### 2.7. Statistical analyses

**2.7.1. Diameter and height.** The overall diameter growth response and height were analysed from the start to the end of experiment using the R System for Statistical Computing (R Core Team, 2023). The relative increment for diameter and height was calculated by dividing the end value by the starting value (Diameter 2/Diameter 1 for diameter, Height 2/Height 1 for height). An initial analysis of variance was conducted on diameter and height to identify any significant differences. Subsequently, a *t* test was employed to compare periodically and continuously irrigated treatments, aiming to ascertain further distinctions. The stem diameter and tree height data for sample seedlings follow normal distribution.

**2.7.2. Leaf water potential.** To examine the relationship between the periodic cycles and LWP during the treatments, a generalised linear model (GLM) with a Gamma distribution and identity link function was used (LWP ~Treatment + Cycle, family = Gamma (link = 'identity'), data = MPa data). Treatment and Cycle were included as predictor variables in the GLM model. We estimated the coefficients for each predictor variable along with their standard errors and corresponding p-values to assess the significance of their effect on MPa. All statistical analyses were performed using R software version 4.3.2, and the significance levels were set at $p < 0.05$.

**2.7.3. Functional xylem traits.** The exported anatomical data were then analysed in conjunction with the experimental growth data in R to obtain growth over time and under the varying environmental conditions. In summary, the fibre, vessel and cell dynamics scripts (see Data Availability) were used to process and analyse anatomical and growth data obtained from the experimental trial. Each script began by loading and preprocessing data, including temperature, humidity, diameter, LWP, wood anatomy and cambial activity. Growth data were then synchronised with physiological metrics, and anatomical data (fibres or vessels) were then filtered and aligned with key experimental periods. Statistical analyses using linear mixed-effects models implemented as (Cell property ~Treatment × Period, random = list (SampleID = ~1, Cell number = ~1), data = cell summary data) in the nlme package of R (R Core Team, 2023) were used to assess differences between the continuously irrigated control and the periodically irrigated treatment, and the day since start of the experiment using the emmeans package.

Treatment and day were specified as fixed effects and sample number as a random effect. This approach was taken for vessel cross-sectional area (CSA), vessel frequency, fibre radial diameter and fibre wall thickness. Vessel frequency overall was taken by counting the number of vessels in each sampled image from the point at which the experiment began until the end and dividing it by the amount of growth in that full period X 0.4 mm (the width of the image in each case). Cambial cells were characterised by thin primary cell walls (Güney et al., 2015; Rossi et al., 2006). Radial number cells in the cambium were counted along three radial files following (Andrianantenaina et al., 2019). Estimation of developmental rates and durations of fibre production and enlargement were calculated following the methods of Drew and Pammenter (2007) (Equations 1–4):

$$\phi i = \frac{g_i}{D_i t_i} \quad (1)$$

where $\phi_i$ is the rate of cell production (cells/day), $g_i$ is the rate of growth in period $i$, $D_i$ is the mean diameter of cells formed in period I and $t_i$ is the number of days in period $i$.

$$t\phi = \frac{n_c}{\phi_i} \quad (2)$$

where $t_\phi$ is the duration of the cell cycle, $n_c$ is the number of cells in cambial zone and $\phi_i$ is the rate of cell production from Equation (1)).

$$t_\sigma \doteq \frac{n_x}{\phi_i} \quad (3)$$

where $t_\sigma$ is the duration of fibre enlargement or wall thickening, $n_x$ is the number of fibres in the enlargement of wall thickening phase and $\phi_i$ is the rate of cell production from Equation (1).

$$\sigma_i = \frac{Size_{end} - Size_{begin}}{t_\sigma} \quad (4)$$

where $Size_{end}$ is the final size of the cells, $Size_{begin}$ is the initial size of the cells and $t_\sigma$ is the duration from Equation (3) and overall calculates the rate of enlargement or wall thickening.

A mixed-effects model, specified again with treatment and time as fixed effects, and sample as a random effect, was used to analyse these variables.

# 3. Results

## 3.1. Environmental and drought-stress conditions

The experiment ran from late March (early autumn) to June (early winter), during which time the temperature averaged 21.8°C and relative humidity about 58%. The highest maximum temperatures during the experiment were recorded in both early and late April during DT1 and WT3. The coolest stages, on average, were WT1 and WT4, although the lowest mean minimum temperature was recorded in DT5.

As expected, the pre-dawn LWP in the CI (control) trees remained constant (Figure 3) while the PI treatment exhibited clear fluctuations, reaching levels as low as −2.4 MPa (Figure 3).

## 3.2. Stem and height growth

Initially, no significant difference in stem diameter was observed between the PI treatment and the CI control trees (Table 3). However, at the end of the experiment, the mean diameter of the CI trees was 0.15 mm greater (p < 0.001) than that of the PI trees (Table 3). Similarly, the initial height of the PI trees was not significantly different from that of the CI trees (Table 3), but by the end of the study, the CI trees were significantly taller (Table 3).

In only two of the drought cycles was a marked shrinkage evident in the PI trees: DT1 and DT5 (Figure 4), although a small shrinkage was discernible at the end of DT3 and DT4. Cycles DT2 and DT6 showed no signs of shrinkage. Some very small shrinkage was observed on occasion in the CI trees, although there was no imposed drought at these times. After water application, there was an initial sharp increase in diameter in PI trees after all droughts, except WT2 and WT6. In these two cases, the level of drought response in the PI trees was not clearly discernible, and the overall rate of growth during DT2 and DT6 showed no discernible decrease in diameter from the CI trees.

Both CI and PI treatments showed a slight decrease (−0.14 mm) in diameter growth rates near the end of the experiment (Figure 4). This decrease was also observed in DT3 in PI trees. During DT4, there was an interesting discrepancy in which the CI trees shrank (despite being continuously irrigated), whereas the PI trees did not (Figure 4). The PI trees, on the other hand, shrunk slightly in the previous measurement. Overall growth was still higher during DT4 in the CI trees than in the PI trees. In treatment DT5 (Figure 4), trees displayed further shrinkage in diameter in the first few days of the treatment (−0.07 mm) as well as towards the end of the treatment (−0.18 mm).

## 3.3. Radial variation in vessel properties

**3.3.1. Vessel area.** There was a significant effect (p < 0.001) of time as well as a significant (p = 0.005) interaction between the treatments and time (i.e., day or period of wood formation) (Figure 5). Overall, the mean vessel CSA increased significantly (p < 0.001) over the course of the experiment in the CI trees at a rate of about 13 $\mu m^2$ per day.

Two periods stood out during which the vessel CSA was significantly (p < 0.05) different between the PI and CI trees. The first was between days 26 and 30, and the second was between days 64 and 70 (Figure 5). These were both periods in which the PI trees were droughted. They were also both periods following about 1 week after the PI trees had reached pre-dawn LWP levels of lower than −2.0 MPa. The PI trees also, when considered relative to the pattern of change in the CI control trees, invariably exhibited a trend of

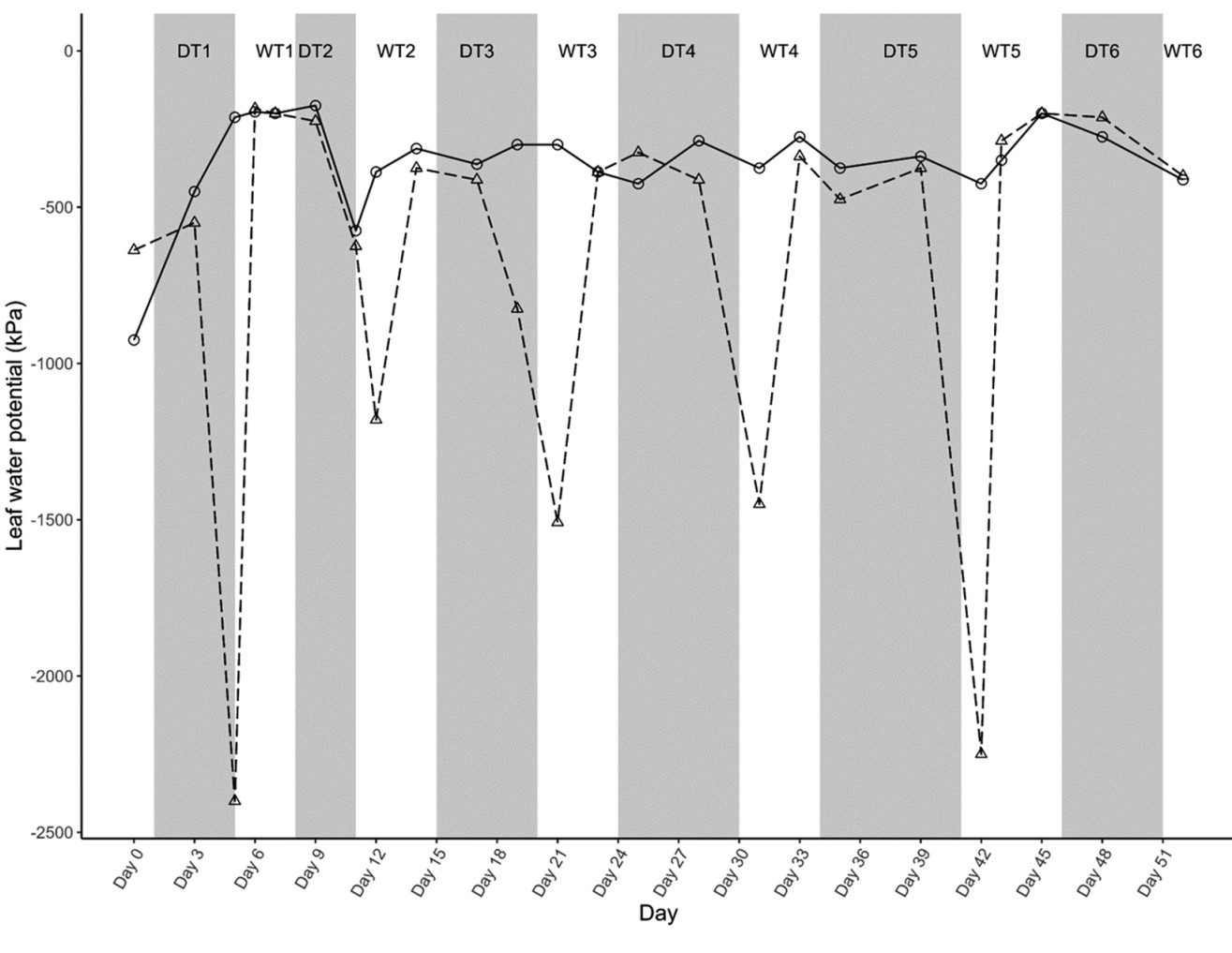

**Figure 3.** Average leaf water potential (MPa) throughout the cyclic drought events. Seedlings that were under drought cycles were periodically irrigated (PI; dashed lines and open triangles) and control seedlings were continuously irrigated (CI; solid lines and open circles).

**Table 3.** Mean stem diameter and height (n = 240) of periodically irrigated (PI) trees and continuously irrigated (CI) trees at the start and end of the experiment

|  | Periodically irrigated | | Continuously irrigated | | T test |
| --- | --- | --- | --- | --- | --- |
| Variable | Mean (mm) | Standard error | Mean (mm) | Standard error | P-value |
| Diameter-start | 3.90 | 0.11 | 4.09 | 0.11 | 0.256 |
| Diameter-end | 5.51 | 0.14 | 6.42 | 0.15 | <0.001 |
| Diameter increment | 1.41 | 0.03 | 1.58 | 0.03 | 0.002 |
| Height-start | 55.63 | 1.98 | 52.55 | 2.05 | 0.287 |
| Height-end | 67.52 | 1.86 | 76.62 | 1.97 | 0.001 |
| Height increment | 1.21 | 0.02 | 1.49 | 0.04 | <0.001 |

decreasing vessel CSA between the end of each droughted period and the end of the subsequent re-watered period, and exactly the opposite pattern in most cases between the end of the re-watered periods and the start of the subsequent drought period (Figure 5). It is notable that during the second last droughted period, almost no growth occurred in most trees and, as a result, very few vessels were found to be allocated to this period.

**3.3.2. Vessel frequency.** On average, the vessel frequency of wood was 138 mm$^2$. There was, however, no evidence of an effect of day (i.e., time into the experiment or the sequence of drought/re-watering cycles) or of any interaction between treatment and day on vessel frequency (Figure 6).

### 3.4. Radial variation in fibre properties

**3.4.1. Fibre CSA.** There was a significant effect of time (p < 0.001) and a significant interaction between treatment and time (p = 0.047). There was a significant (p < 0.001) overall decrease in fibre area over the course of the experiment in the PI trees, with a 1.7 μm$^2$ change

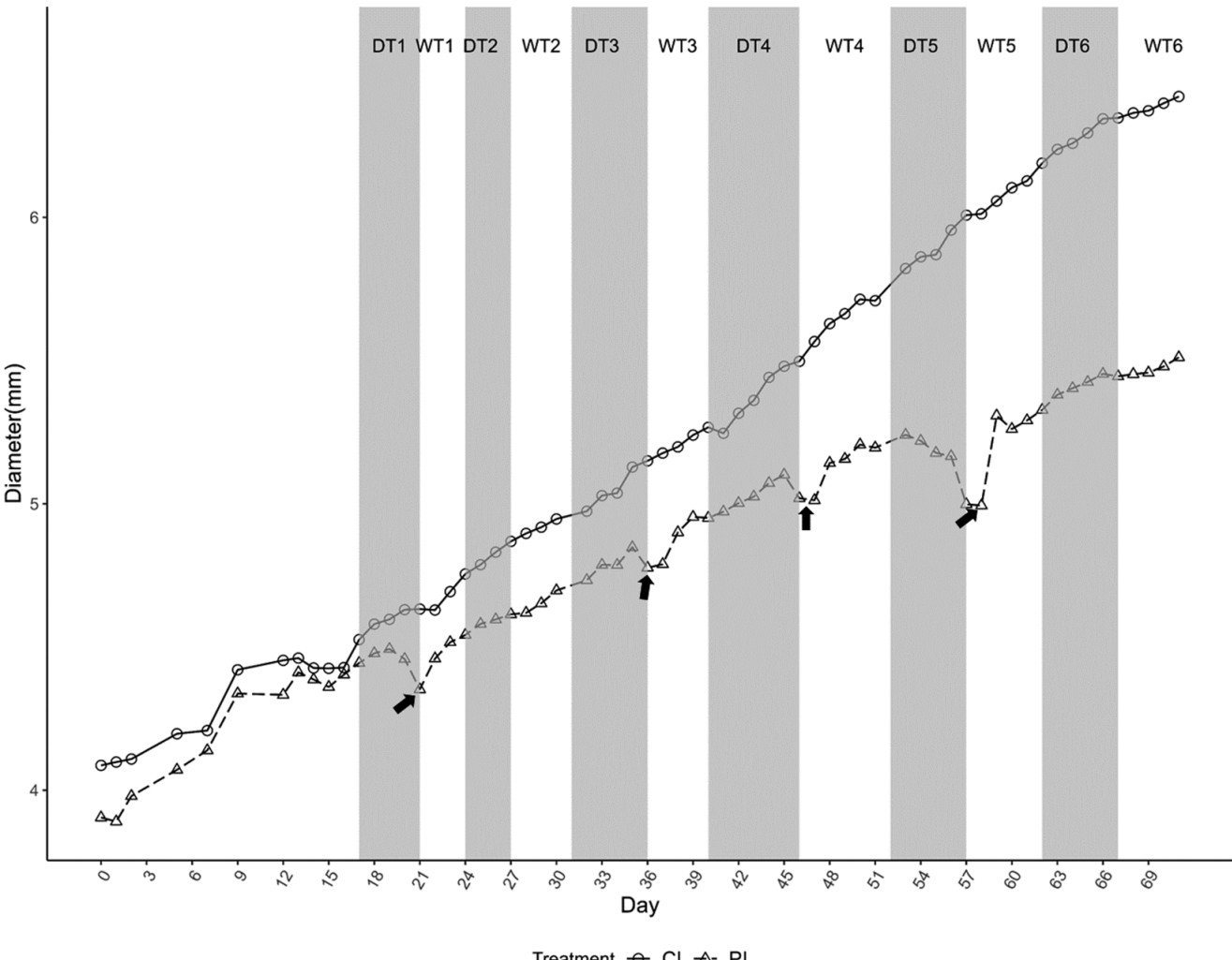

**Figure 4.** Growth response of *E. cladocalyx* when seedlings were periodically irrigated (PI; open circles) and continuously irrigated (CI; open triangles) throughout the experimental trial. PI seedlings were droughted until signs of visible stress (drought treatment, DT) followed by watering them up to field capacity daily (watering treatment, WT) for six periods. Shrinkage represented by arrows.

per day, on average. Over the periods of days 23–28 and days 64–71, there were significant (p = 0.05 and p = 0.03, respectively) differences in the fibre CS area between the PI and CI trees (Figure 7). In the PI trees, there was generally a significant (p < 0.001) difference between the areas of fibres formed after day 60 and those formed before about day 50 (Figure 7). This corresponds to the period before and after the drought period, which temporarily stopped growth in the PI trees.

**3.4.2. Fibre cell wall thickness.** There was a significant (p < 0.001) effect of time on fibre wall thickness variability and a significant (p < 0.001) interaction between time and treatment. Fibre WT was significantly (p = 0.01) smaller in the PI trees compared to the CI trees between days 18 and 24 (the first drought period) (Figure 8) and significantly larger between days 24 and 28 (after re-watering) (Figure 8). The same was true between days 60–65 and 65–72, with the WT lower in the period of drought in the PI trees and higher in the watered period in the PI compared to the CI trees (Figure 8). In both cases, these differences followed the two most extreme droughting events. There was generally a pattern of wall thickness declining following re-watering, although this was variable across the study. The period from about days 64 to 72 (Figure 8) emerged

as being significantly (p < 0.05) different in the PI trees compared to other periods, with much smaller mean wall thickness.

### 3.5. Xylem and cambial variation between the treatments

Microscopic investigation of vessel and fibre properties in periodically irrigated seedlings during days 60–72 (Figure 9, blue block) showed relatively small, packed fibres that were not obviously thick-walled. During this period, multiple solitary, small- to medium-sized vessels were generally formed. Fibres generated between days 18 and 28 (Figure 9, black block) were somewhat larger. What was clear, however, was the level of variability between samples. Some samples produced much larger fibres, or potentially more vessel-associated tracheids and parenchyma. In addition, in the period between days 18 and 28, it was interesting to note that in some samples almost no vessels formed, while in others several large, fully formed vessels were evident (Figure 9). This same kind of variability was also observed in the control (continuously irrigated) samples (Figure 10) over that time.

Surprisingly, there was overall no significant different in the number of cells in the cambial zone between the control and PI treatment with a mean for the full dataset of about 3.9 ± 0.3 cells

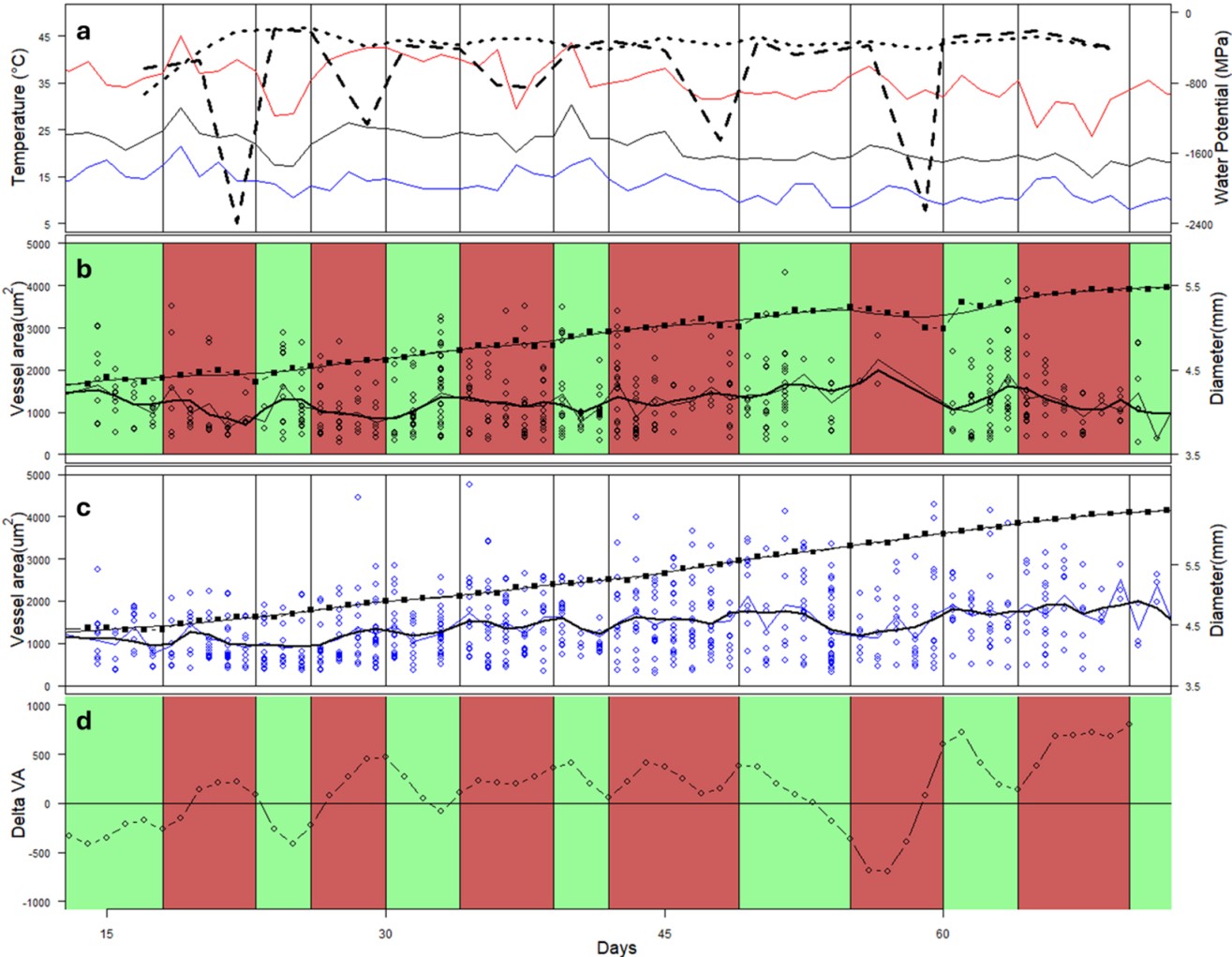

**Figure 5.** Variation in environmental conditions, stem diameter and vessel cross-sectional area in CI and PI seedlings over the course of the 75-day experiment. Mean minimum and maximum temperatures are shown as black, blue and red lines, respectively, with mean pre-dawn leaf water potential (CI = dotted; PI = dashed) (a). Timing of drought/watering cycles for the PI trees is shown in green (watered) and red (droughted) (b). Mean growth increments over 2- to 3-day periods are shown as closed squares with bold line, and areas of individual vessels in each growth increment are shown as open dots (b, c). A smoothed line shows the trend of vessel area. The difference between the smoothed trend of VA in the PI and CI trees is shown in (d).

wide. The cambial zone was, however, evidently narrower and less active during days 18–28 than in days 60–72 (Figure 9) in the PI trees. This was not clearly the case for the control trees. There was also generally less evidence of incipient vessels (newly forming adjacent to the cambial zone) during days 18–28 compared to days 60–72.

Overall, the rate of fibre enlargement did not differ between treatments, but the duration of the period of fibre expansion was found to be significantly (p = 0.030) longer in the periodically irrigated trees. The duration of cell enlargement was longer and the rate of enlargement was lower, during the in-between period and in Period 2 compared to Period 1 (Table 4). This was true of both the periodically droughted treatment and the control. The rate of fibre production did not differ between treatments or the growing periods. The duration of the cell cycle was significantly (p = 0.040) longer in periodically irrigated trees compared to the control. There was also notable variation with considerable increase occurring both between Periods 2 and 1 and in the in-between period and Period 1 for the duration of the cell cycle.

Also notable in the micrographs, although not an effect caused during the period of our study, was the marked and sudden

transition at an earlier stage in the life of the seedlings from markedly thick-walled cells to generally larger, thinner-walled cells (arrows and Figures 9 and 10). This corresponded, likely, to a spring growth flush in about September 2016.

## 4. Discussion

### 4.1. Growth response

We studied short-term growth response and xylem anatomy response in *E. cladocalyx* seedlings exposed to cyclic versus consistent watering regimes. The drought cycles led to periodic and marked reductions in LWP that impacted on overall diameter and height growth compared to the control trees. This was in agreement with comparative studies on drought-tolerant eucalypts, which have demonstrated differences in water stress responses, showing that seedlings such as *E. cladocalyx* have adapted to dry conditions and developed responsive mechanisms to withstand both short-term droughts (Mora et al., 2009; Akhter et al., 2005). However, this study also highlighted the inconsistency in the droughting events as responses to drought were only discernible in two of the

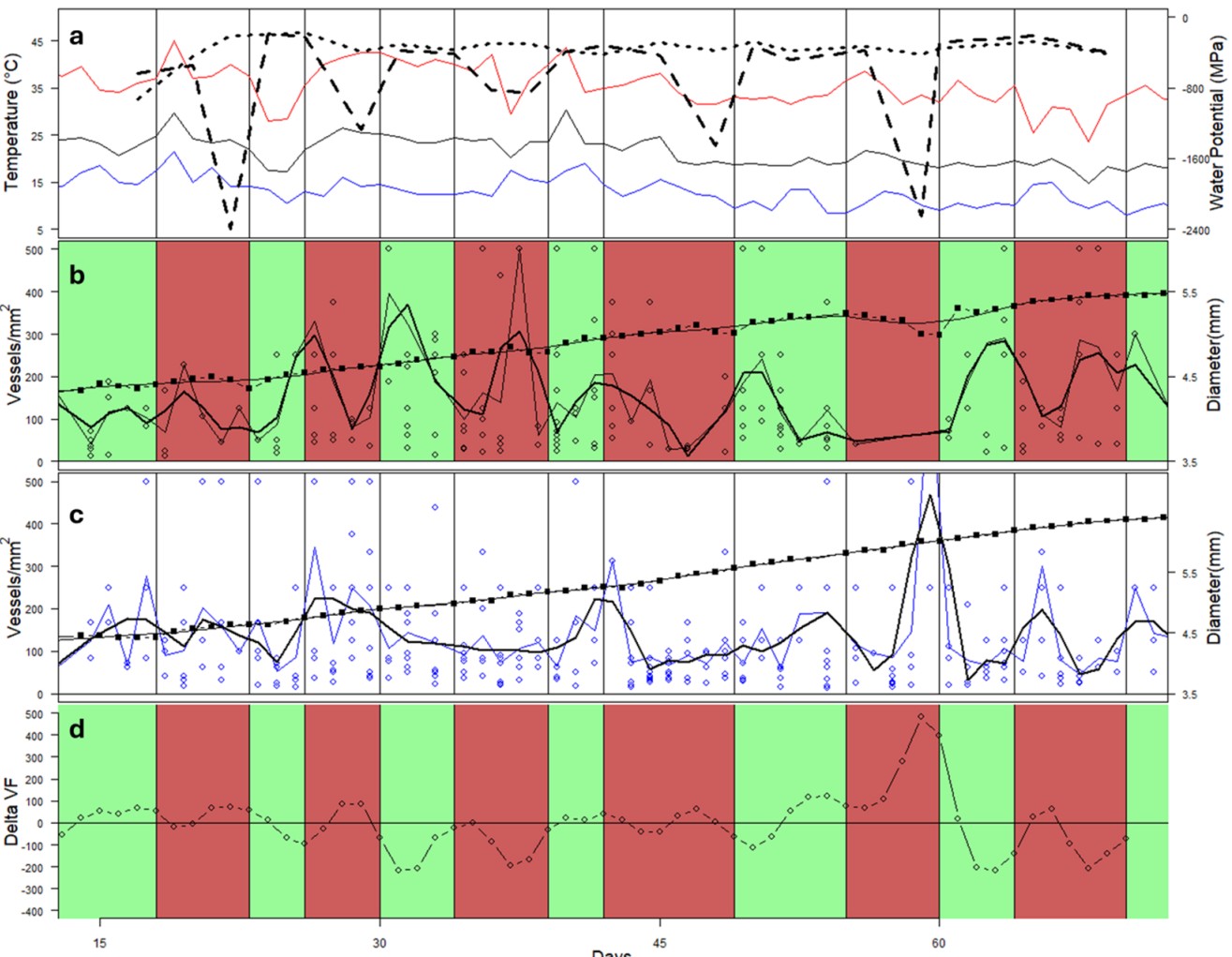

**Figure 6.** Variation in environmental conditions, stem diameter and vessel frequency in CI and PI seedlings over the course of the 75-day experiment. Mean minimum and maximum temperatures are shown as black, blue and red lines, respectively, with mean pre-dawn leaf water potential (CI = dotted; PI = dashed) (a). Timing of drought/watering cycles for the PI trees is shown in green (watered) and red (droughted) (b). Mean growth increments over 2- to 3-day periods are shown as closed squares with bold line, and vessel frequency in each growth increment are shown as open dots (b, c). A smoothed line shows the trend of vessel area. The difference between the smoothed trend of VF in the PI and CI trees is shown in (d).

six drought events (DT1 and DT5). The continuous exposure to the cyclic drought treatments is likely to have led to greater tolerance (Pritzkow et al., 2021; Hodecker et al., 2018; Guarnaschelli et al., 2003) and improve survival during subsequent drought events (Anderegg et al., 2013; Guarnaschelli et al., 2006; Searson et al., 2004). Therefore, eucalypts show acclimation when exposed to drought overtime and this has been established in *Eucalyptus* studies (Chemlali et al., 2022; Saadaoui et al., 2017; Garau et al., 2008; Li et al., 2000).

We were interested, also, in how easily the final xylem properties could be linked to cambial zone dynamics during the experiment. Surprisingly, we did not find a clear link and differences in the number of cambial cells between the treatments (PI vs. CI) to be as clear as expected. On the other hand, the widths of the zones differed. In this context, it is notable that studies have shown that seedlings maintained intact cambial cells during drought stress, demonstrating the cambium's responsiveness and resistance to drought (Li & Jansen, 2017). However, it has been debated whether this resilience extends to both droughted and sensitive species (Begum et al., 2013). In concert with these findings, there was also a surprisingly small difference overall between the rates of cell production

(according to our methodology). There was also no significant effect of the drought treatment (overall) on the duration of the cell cycle. However, there was evidently a general change that occurred over the course of the study, which was seen in both treatments.

## 4.2. Xylem properties variation

**4.2.1. Vessel area.** The decrease in mean CSA in vessels over some periods in this study suggests that hydraulic adjustments occurred in relation to periodic droughting (Chambi-Legoas et al., 2023; February et al., 1995). Other studies have highlighted that vessel size can be influenced by a severe decrease in diameter growth (Ohashi et al., 2015). Water limitation is generally known to reduce vessel lumen area (Chambi-Legoas et al., 2023; Huang et al., 2022; Câmara et al., 2020; Searson et al., 2004) likely via reductions in achievable turgor pressure (Ohashi et al., 2015). This was also in agreement with studies done on other eucalypts, such as *Eucalyptus marginata*, that showed the formation of frequent and narrow vessels during drought, and how this was accompanied by a reduction in vessel lumen area (Huang et al., 2022; Searson et al., 2004).

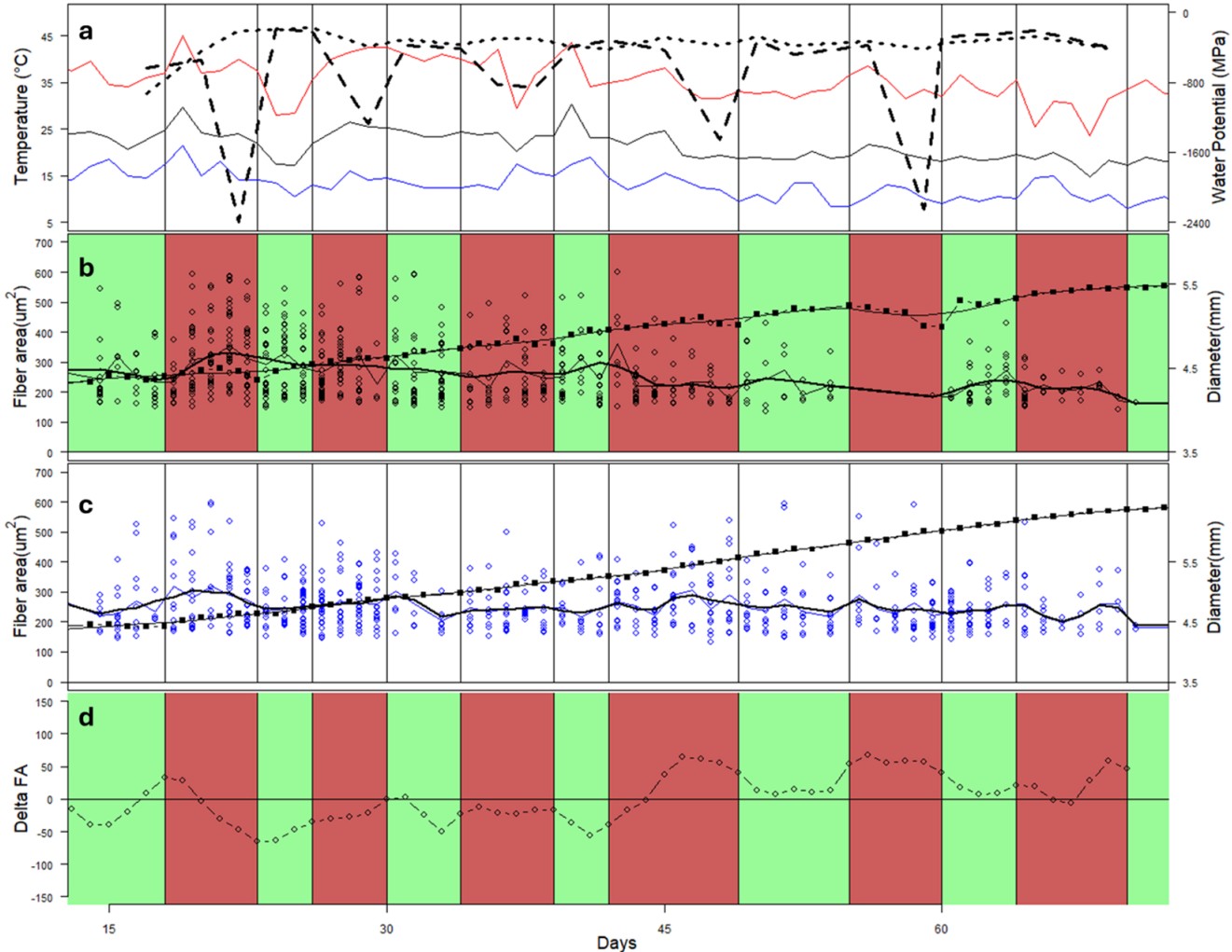

**Figure 7.** Variation in environmental conditions, stem diameter and fibre cross-sectional area in CI and PI seedlings over the course of the 75-day experiment. Mean minimum and maximum temperatures are shown as black, blue and red lines, respectively, with mean pre-dawn leaf water potential (CI = dotted; PI = dashed) (a). Timing of drought/watering cycles for the PI trees is shown in green (watered) and red (droughted) (b). Mean growth increments over 2- to 3-day periods are shown as closed squares with bold line, and fibre cross-sectional area in each growth increment is shown as open dots (b, c). A smoothed line shows the trend of fibre cross-sectional area. The difference between the smoothed trend of VF in the PI and CI trees is shown in (d).

Studies on *Eucalyptus* have shown that the average diameter of vessels in *Eucalyptus* wood is significantly influenced by water availability (Barbosa et al., 2019). The vessels of the PI treatment were not overall significantly different in size, but only for two periods when the trees experienced the imposed drought most intensely, with pre-dawn water potential less than about −2 MPa. This strongly suggests that only above a threshold of drought exposure (Pritzkow et al., 2020), for these seedlings, did vessel expansion begin to be affected, this also agreed with studies that showed that anatomical changes are based on the intensity and duration of environmental conditions (Chambi-Legoas et al., 2023). Other studies have found that water limitation significantly reduced mean vessel lumen area in species such as *E. marginata* (Searson et al., 2004).

**4.2.2. Vessel frequency.** It was surprising that the differences in vessel frequency between control and droughted trees in our study were not more distinct, given the finding of drought on this parameter by other authors (Ohashi et al., 2015; Arend & Fromm, 2007; Mauseth & Stevenson, 2004; Searson et al., 2004; Schume et al.,

2004; Carlquist, 1966; February & Manders, 1999). It was also surprising that vessel frequency did not show clearer adjustments during drought periods. It is, however, notable that other *Eucalyptus* studies have also not found that vessel frequency changes with environmental conditions (Freitas et al., 2019; Salvo et al., 2017). In general, changes in vessel size (which were observed) are linked to changes in vessel frequency, and studies have found that the balance between vessel size and frequency is a critical adaptation to changing environmental conditions (Barbosa et al., 2019). However, perhaps, in this case, the relatively brief periods between drought events were too short for a clear vessel frequency change to be observed. The resolution of our methodology may also have meant these changes were too ambiguous.

**4.2.3. Fibre area.** Changes in fibre features were observed particularly in two periods in this study. In both cases, the effects followed periods of particularly levels of stress (pre-dawn water potential < −2 MPa). This was, like for vessels, evidently a threshold below which expansion processes and thus total size became critically reduced (Barbosa et al., 2019). Water limitation, especially under

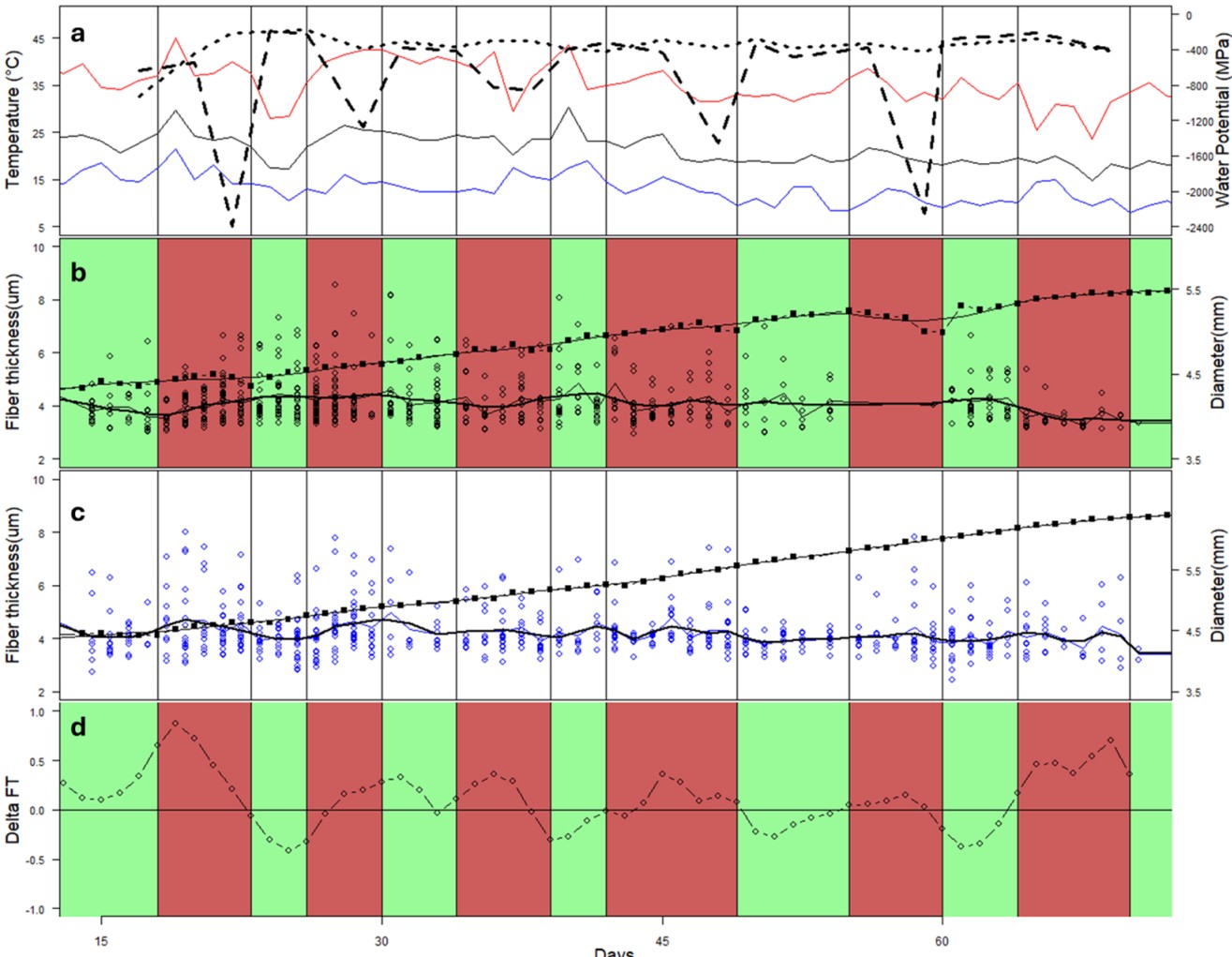

**Figure 8.** Variation in environmental conditions, stem diameter and fibre wall thickness in CI and PI seedlings over the course of the 75-day experiment. Mean minimum and maximum temperatures are shown as black, blue and red lines, respectively, with mean pre-dawn leaf water potential (CI = dotted; PI = dashed) (a). Timing of drought/watering cycles for the PI trees is shown in green (watered) and red (droughted) (b). Mean growth increments over 2- to 3-day periods are shown as closed squares with bold line, and fibre wall thickness in each growth increment are shown as open dots (b, c). A smoothed line shows the trend of fibre wall thickness. The difference between the smoothed trend of VF in the PI and CI trees is shown in (d).

high temperatures, can generally be expected to lead to a reduction in fibre cross-sectional dimensions (Barbosa et al., 2019; Salvo et al., 2017; Drew & Pammenter, 2007; Leal et al., 2003). Our finding did, however contrast with some research that has shown an increase in fibre area during drought (Moulin et al., 2022). It has also been found that fibre size in *Eucalyptus* species tend to be less responsive under changing environmental conditions (Chambi-Legoas et al., 2023; Câmara et al., 2020). The effect of vessel-associated tracheids may also have been a factor in our data, but it was not explicitly considered (Moulin et al., 2022).

Differences in fibre areas were not evidently caused by differences between the treatments in the rate of the expansion process. However, potentially, the differences we observed in the estimated duration of enlargement may be important. The longer duration may imply the need to hold cells in this developmental stage for longer to achieve a critical size, but under conditions of more limiting negative pressures where turgor is harder to maintain.

**4.2.4. Fibre wall thickness.** Fibre wall thickness showed more fluctuations in periodically irrigated seedlings. This was observed in more than two periods throughout the trial. Due to this response, it was expected that fibre thickness would be higher in periodically irrigated seedlings versus the control. However, there was no significance between the two treatments. To alleviate negative hydraulic adjustments/pressure, some studies have suggested that thicker walls are produced within the period of drought (Santos et al., 2021; Barbosa et al., 2019), which was not necessarily the case in this study. Our findings were in contrast with studies that found that *Eucalyptus* species produce thicker fibre walls as a strategy to deal with water stress during drought (Moulin et al., 2022; Santos et al., 2021; Barbosa et al., 2019; Whitehead & Beadle, 2004), as well as increase survival of upcoming drought conditions (Moulin et al., 2022). This response is not necessarily a universal pattern for *Eucalyptus* (Barbosa et al., 2019), and the lack of variation in fibre cell wall thickness between PI and CI in *E. cladocalyx* agrees with studies that found no differences in cell wall thickness in species such as *Eucalyptus obliqua* (Searson et al., 2004), which also consisted of trees that were previously exposed to drought and well-watered conditions. Like fibre area, fibre thickness in *Eucalyptus* species was found to be less responsive under environmental

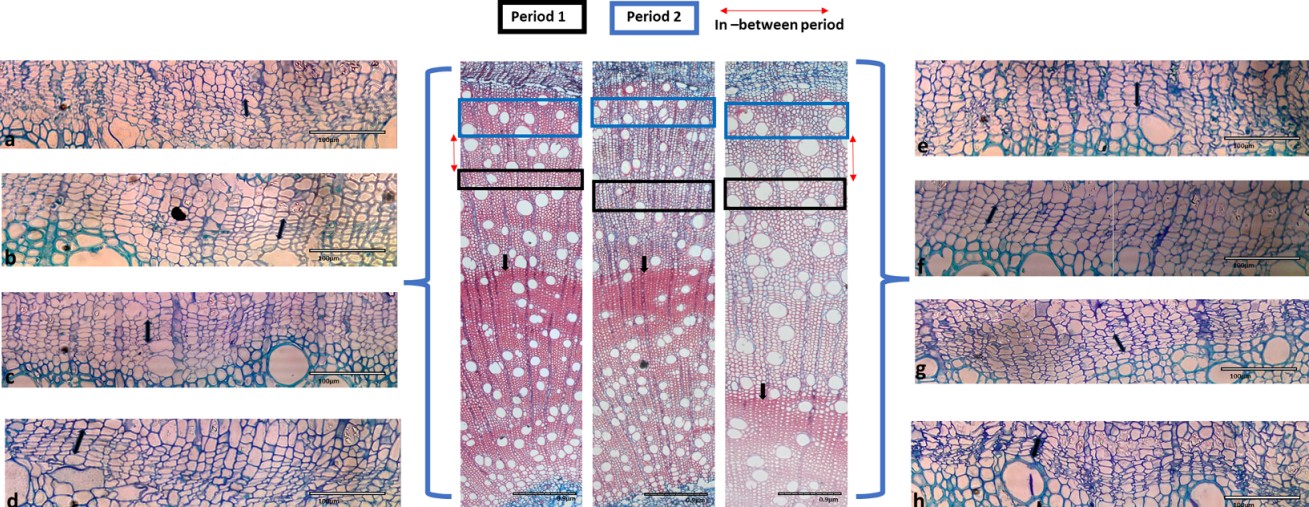

**Figure 9.** Microscopic image of 1.5-year-old *Eucalyptus cladocalyx* seedlings illustrating radial variation in vessel and fibre properties observed in PI treatment over the 75-day experiments. Significant differences in vessel and fibre features were observed during days 18–28 (Period 1, black square) and days 60–72 (Period 2, blue square). Red arrow indicates the period in between Periods 1 and 2. Black arrows illustrate the change in fibre thickness before the start of the experiment. Cambial images taken during the experimental trial are shown on the right and left sides, showing the activity of the cambium during PI events. Images (a) and (b) show the zone during days 60–72, images (c–f) show the zone during days 29–59 and images (g) and (h) show the zone during days 18–28. PI seedlings were droughted until signs of visible stress (drought treatment, DT) followed by daily watering to field capacity (watering treatment, WT) for six periods. Periods of growth were calculated using the change in diameter and the resulting cumulative sum widths of diameter in conjunction with the y-centroid values to create the different zones.

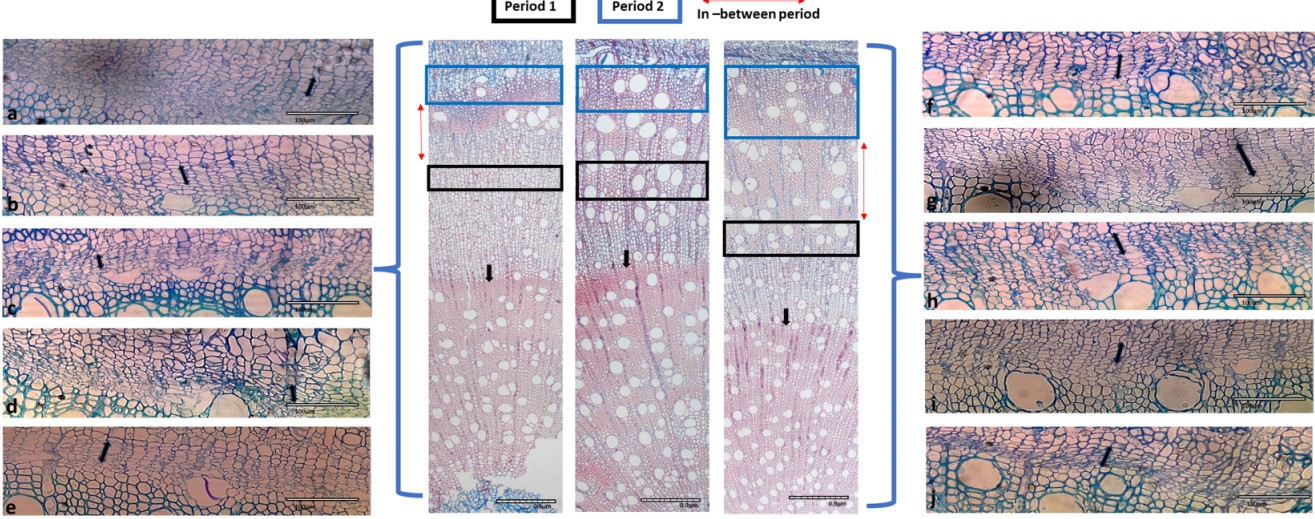

**Figure 10.** Microscopic image of 1.5-year-old *Eucalyptus cladocalyx* seedlings illustrating radial variation in vessel and fibre properties observed in CI treatment over the 75-day experiments. Significant differences in vessel and fibre features were observed during days 18–28 (Period 1, black square) and days 60–72 (Period 2, blue square). Red arrow indicates the period in between Periods 1 and 2. Black arrows illustrate the change in fibre thickness before the start of the experiment. Cambial images taken during the experimental trial are shown on the right and left sides, showing the activity of the cambium during control events. Images (a) and (b) show the zone during days 65–72, (c, d) show the zone during days 51–64, images (e) and (f) show the zone during days 46–50, images (g) and (h) show the zone during days 36–45 and images (i) and (j) show the zone during days 27–35. The CI seedlings were continuously watered throughout the experiment as a control (C). Periods of growth were calculated using the change in diameter and the resulting cumulative sum widths of diameter in conjunction with the y-centroid values to create the different zones.

**Table 4.** Results of mixed-effects model developmental rate and durations of two xylem growth periods (Period 1 (days 18–28) and Period 2 (days 60–72)) during periodically irrigated (PI) and continuously irrigated (CI) treatments

| Parameter | Period 1 | | In-between period | | Period 2 | |
|---|---|---|---|---|---|---|
| | Control (continuous) | Periodic | Control (continuous) | Periodic | Control (continuous) | Periodic |
| Rate of fibre production | 2.0$^B$ | 2.2$^B$ | 2.2$^B$ | 1.3$^A$ | 1.6$^A$ | 1.6$^A$ |
| Duration of cell cycle | 1.8$^A$ | 2.0$^A$ | 2.5$^B$ | 3.6$^B$ | 2.6$^B$ | 2.6$^B$ |
| Duration of enlargement | 2.2$^A$ | 2.2$^A$ | 2.5$^B$ | 3.7$^B$ | 3.1$^B$ | 3.6$^B$ |
| Rate of enlargement | 8.4$^B$ | 9.4$^B$ | 7.6$^A$ | 5.1$^A$ | 5.2$^A$ | 4.3$^A$ |

*Note*: Differing capital letters show significant differences at alpha = 0.05 between periods for each treatment.

changes (Chambi-Legoas et al., 2023; Câmara et al., 2020; Freitas et al., 2019).

## 5. Conclusion

The growth response of *E. Cladocalyx* changed progressively during the duration of the drought, and seedlings showed signs of recovery when watered again. Similarly, cellular modifications appeared to shift in response to growth when seedlings were subjected to cyclic drought. However, there were no significant differences in vessel area, frequency, fibre area or fibre thickness between periodic and continuous irrigations. demonstrating the growth response of *E. Cladocalyx* can decrease, but its cellular characteristics may take longer to reflect this change, which may be impacted by the intensity of the drought, which in our study was not severe enough, or by the different reactions of *Eucalyptus* species. On the other hand, the response of increased water availability does not necessarily equate to lower vessel frequency and increased diameter (Barbosa et al., 2019; Lundqvist et al., 2017). In addition, some studies have noted that responses and changes may not be present all the time when plants are exposed to repeated drought exposure (Pritzkow et al., 2020). Long-term drought response may differ from short-term exposure (Zhou et al., 2016), and the difference in drought stress may be due to the duration of drought stress; age becomes a contributing factor to the differences in drought treatment (Zhou et al., 2016). It becomes clear that changes in *Eucalyptus* influenced by environmental changes tend to be overtaken by the genetic diversity (Wilkes, 1988) or other factors.

**Supplementary material.** The supplementary material for this article can be found at http://doi.org/10.1017/qpb.2025.7.

**Data availability statement.** The data that support the findings of this study are openly available in Zenodo at http://doi.org/10.5281/zenodo.13799117.

## Acknowledgements

The authors would like to thank Francis Bouwer for conducting the experiment and collecting data and Leandra Moller and Letitia Schoeman for technical assistance, helpful comments and proofreading the article. Ayodeji Oyedeji did the sectioning and resin mounting of cambial samples.

**Funding statement.** This work was supported by the Hans Merensky Legacy Foundation (S005986) and Stellenbosch University through the subcommittee B funding instrument.

**Competing interest.** The authors declare no competing interests.

**Open peer review.** To view the open peer review materials for this article, please visit http://doi.org/10.1017/qpb.2025.7.

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
