## [Reviewer Report]

Comments

This manuscript describes the effects of drought on wood formation in Eucalyptus cladocalyx. The manuscript is well organized. My comments are as follows:

1) L91: Where is the answer to question 3b?

2) L98-99: Authors should include more detailed information about seedlings. When we evaluate the effect of environmental factors on wood properties, we should use clonal materials. Instead of clonal materials, the authors used seedlings. Thus, more detailed genetic information about the seedlings is needed. In addition, stem diameter and tree height data for sample seedlings follow normal distribution. This information is also essential.

3) L105: Is the greenhouse located in the same place as the GPS code presented in L99? If not, please include the greenhouse GPS code information. If we find the information, we can imagine the climatic conditions, such as the photoperiod.

4) L130-131: The authors measured the stem diameter 2 cm above the ground using a caliper. Is the stem circular? The stem near the base sometimes shows a circular shape.

5) L201-207: The authors should also present the model form by formula as well as text. This is the same in the next section (Functional xylem traits).

6) L240-241: The authors used a greenhouse. However, the experiments might have influenced the outside temperature. Thus, the authors should present outside climatic data (at least temperature), even in supplemental materials.

7) L245 (Table 1): Please use uniform significant digits.

8) L266: Figure use unit of kPa. Please use a uniform unit.

9) L283-289: Where is data? It is tough to read the data from Figure 4.

10) L294-297 (Table 3): Please provide the number of samples.

11) L300-: Even in supplemental materials, statistical values of measured anatomical characteristics are needed.

12) Figure 2: Is it needed? If the authors need it, please include related information in this figure.

13) Figures 3 and 4: Please include cycle information in the figure (like figure 5 and others).

14) Figure 4: Please provide some arrows for indicating shrinking phenomena.

15) Figures 9 and 10: More magnified photomicrographs are needed. It is tough to recognize.

---

## [Reviewer Report]

The paper QPB-2024-0070 presents the effects of repeated periodic droughts on xylem features (vessel and fibre sizes, vessel density, fibre wall thickness) and cambial activity in the supposedly drought-adapted Eucalyptus cladocalyx. Experiments were performed using less than one-year old seedlings subjected to six repeated cycles of water deficit. Measurements relied on xylem histology and microscopic evaluations of stem cross-sections. The authors report highly variable complex patterns depending on the period considered, most likely because of differences in stress intensity and intrinsic genetics.

I believe the paper fits the journal scope and is of potential interest to the reader. The paper clearly has merit by including data from 240 individual seedlings obtained after histological preparation and image analysis for several xylem cell features. However, I see a couple of issues that, so far, prevent the paper from acceptance in my opinion.

1. I did not see why the authors chose to apply such a complex experimental design with six repeated drought cycles rather than a controlled long-lasting moderate drought. Unless I am wrong, I did not see a clear rationale arising from the literature or from hypotheses. I believe such numerous, short and repeated drought events make the histological characterization highly complex so that to identify the exact periods/xylem during which cells were produced (see comment below). It also questions the legacy effects of past drought cycles on xylem cell features as the experiment advances, and this is confounded with cambial seasonal aging.

2. In line with the comment above. Unless I am wrong, I did not see any explanation regarding how measurements of xylem cell features were performed on the trees sampled every three days for examining temporal variations. I mean, what zone between the cambium and the pith was used to make morphometric measurements? Without specific marking or high-resolution growth data, how was this realized to be sure of what was measured was absolutely specific of that period of the experiment with growth constantly varying in response to variable watering treatments? Fig. 9 and 10 kind of show different zones in the xylem corresponding to supposedly different drought cycles, but I could not understand how this was performed either based on the text manuscript or even from the figure legends.

3. Although the authors measured predawn leaf water potential, the implementation of drought in the experiment seems empirical as cycle duration was variable and “adjusted based on visible signs of stress” (see L.119). Which signs? Stress intensity thus also varied from one cycle to the other. The authors actually acknowledge in the conclusion section that differences in stress intensity and duration may explain the variable patterns observed, but this remains speculative.

4. The authors used seedlings which means that each individual was genetically different. Although seeds were obtained from a single stand, genetic diversity remains unknown and I fear that the variable temporal trends obtained from four different seedlings sampled every third day may be partly explained by this issue. In their defence, the authors also acknowledge this issue in the conclusion section (L. 505) but this still raises the question of whether or not the absence of typical patterns for xylem cell features is something “normal” or is masked by the genetic diversity. In addition, from what I understand, morphometric measurements were performed on one single subsample per individual (see L.156, 1/8 of the circular surface). Typically, such measurements are repeated at least two or three times on opposite or distributed radial sectors to account for spatial heterogeneity which can be significant, especially on young seedlings with small stems. This may also explain variation in data.

For these main reasons, I believe the paper cannot be accepted in its present shape. However, the authors may consider a resubmission if they are able to address these issues.

Specifics:

- I found in several instances sentences were too vague. See for instance L.40-42, 59-60 or 76 in the introduction. See also L.396-399 or 406-409 in the discussion.

- L.37: ‘Eucalypts rely heavily on xylem tissue to respond to drought stress’. I believe this can apply to any species. I would remove it.

- L.38-39: ‘as susceptibility to embolism increases’. Pls change by ‘as embolism increases’.

- L.40-43: this deserves to be explained a bit more. What kind of changes?

- L.47: ‘non-uniform xylem developmental patterns’: what is meant here?

- L.52: has instead of have

- L.59-60: what is meant here? Pls specify.

- L.76: better than what?

- L.89-90: hypothesis 3a. I believe this cannot be assessed from the experimental design used in this paper. Potential legacy effects of past drought events accumulate with time such that the effects of multiple repeated droughts and season are confounded.

- L.102-103: the difference in Soil VWC is actually quite important here (1-10%), especially if we consider a sandy soil. Was this meant for control plants before the experiment started? In addition, volumetric soil water content has no real significance per se if the values corresponding to field capacity and permanent wilting are ignored. As such, the 1 to 10% difference could be dramatic in terms of water availability.

- L.138-141: pls indicate the measurements were performed before sunrise.

- L.185: ‘Software and image processing’: I think this part would deserve additional information. Pls also see one of my previous comment regarding the methodology for determining the xylem areas related to each specific drought period.

- L.219: ‘cambial’: cambium? Cambial zone?

- L.269: ‘Overall’: not necessary

- L.283-289: I found this part hard to follow, probably because there is no reference to any figure and the patterns seem very variable. Wonder whether this is necessary.

- Table 3: pls adjust the number of digits for the error related to diameter and height for the PI seedlings.

- L.298 and 326: ‘Radial variations’. Not sure this is actually radial variations that are presented. Once again, from what I understand, these are the temporal trends estimated from punctual individual measurements during the whole experiment, but measurements were systematically made from the same xylem area right below the cambial zone?

- L.320-325: I understand that vessel area might be more precise than vessel diameter, especially if vessels are elliptical. However, I think additional information about corresponding vessel average diameter might be useful just for the sake of comparison and simplicity.

- L.347-348: here and elsewhere, careful, only present significant patterns.

- L.364-365: same comment, avoid ‘weakly significantly’ when the p-value is higher than the cut-off.

- L.396-399: too vague, pls reword.

- L.401 ‘which was the case in our observations’: you cannot write this directly since you did not compare your findings with larger trees.

- L.407-409: too vague, pls reword.

- L.504: becomes

- Fig.2: pls add ‘predawn’ before water potential

- Fig.3: I am not sure this figure is needed since it is somehow replicated in Figs 5, 6, 7 and 8. Also, pls convert kPa in MPa in all figures.

- Fig.4: same comment as for Fig.3. If this one is conserved, it might be worth adding in the back some colour to represent start and end days for all drought cycles (as actually made in Figs 5, 6, 7 and 8).

- Figs 5, 6, 7 and 8: from what I understand, at each date (i.e. every third day), two seedlings per block and per treatment were sampled which makes four seedlings per treatment if I am right. What do individual points in b) and c) panels refer to then? Also, why is the number of points in b) and c) panels for vessel density (Fig.6) less numerous than for other traits (Fig. 5, 7 and 8)?

- Figs 9 and 10: I did not understand how the different periods were delimited (pls see my previous comments). I also do not understand what the three central images represent. What are the arrows for?

---

## [Editor Report]

Thank you your submission to the special collection ‘Advances in xylem and phloem formation research’ of QPB. I have now received the reports of two expert reviewers.

It has been appreciated that your study is based on a large amount of data and analyses. Some important points are raised in the reviewers' reports to further improve and clarify your manuscript. Especially, better explanations are needed on your experimental design and anatomical measurements.

I look forward to receiving your revised manuscript in which you carefully address all points raised by the reviewers.

---

## [Reviewer Report]

Manuscript ID: QPB-2024-0070-R1

I already reviewed the original manuscript (reviewer 2). The authors have reworded a couple of paragraphs, replied to my main comments regarding the use of multiple droughts, treatment implementation and genetic diversity and addressed many of my specific comments. Overall, I can understand the general idea of investigating the drought response in this species and using multiple drought cycles. However, although it does not take away the merit of the work accomplished by the authors, I still believe the findings remain elusive partly because of the limits I raised and as attested by the final conclusion in the abstract “… anatomical properties varied complexly and inconsistently across drought cycles, likely due to differences in drought intensity, strategies and genetic factors”.

Besides, I still had issues understanding Figures 9 and 10 (see my previous comments):

- If I am guessing correctly the different areas corresponding to the different periods were delimited based on growth measurements. Besides methodological limits related to this (growth measures include wood production but also outer tissues), the rather narrow windows on the images suggest it might have been difficult to obtain robust estimations for vessel area or vessel density considering the very few number of vessels included? I also suggest the authors add one or two lines in the figure legend regarding how the frames were delimited.

- What do the three images in the centre represent? Different trees? Unless I am mistaken, this is not specified.

- Why were Period 1 and Period 2 chosen for the illustration? If this is because these correspond to the periods during which trees were the most responsive, this needs to be indicated.

- What are the changes in fibre density highlighted by the black arrows indicative of, if this occurred before starting the experiment?

Unanswered question: Figs 5, 6, 7 and 8: from what I understand, at each date (i.e. every third day), two seedlings per block and per treatment were sampled which makes four seedlings per treatment if I am right. What do individual points in b) and c) panels refer to then? Also, why is the number of points in b) and c) panels for vessel density (Fig.6) less numerous than for other traits (Fig. 5, 7 and 8)?

There are still mistakes regarding the use of pressure units (kPa vs. MPa). Pls convert units into MPa as these are the typical units for water potential and hydraulics. See for instance L. 214, L.280, L.460, etc…

It seems that there is supplementary material associated with the manuscript. However, unless I am wrong, I did not see it called upon or referenced in the text.

Specifics:

- Abstract L.14-15: I would change ‘daily variation’ by ‘temporal variation. Considering the design, deciphering daily changes in xylem cell features seems misleading/unrealistic although I understand the morphological variables are plotted against time.

- Abstract: I would remove the p-values from the abstract, no need here.

- Abstract L.17: I would move this sentence above, to be the first sentence introducing the findings.

- L.42: consider deleting the full stop after (Lens et al. 2022).

- L.47: parenthesis after 2021

- L.62: o is missing in cladocalyx

- L.78-79: be more specific. What vessel and fibre features?

- L.108-111: thanks for adding this this is now clearer.

- L.196: pls delete ‘But’

- L.319: pls remove ‘distinctly’

- L.323: pls add the full meaning for LWP (leaf water potential the first time the abbreviation is used)

- L.331: no need for two digits here after the coma

- L.398-401: I did not understand how this could be directly related to the findings or how this could be ‘in agreement’ since the study referenced here seems to be comparing drought tolerant vs. sensitive material. Pls reword.

- L.403: ‘in two of the six drought events’: which ones? Pls indicate. This is particularly important if we are to consider the next sentences and the assertions regarding acclimation for potentially enhanced tolerance/resilience.

- L.407: pls remove ‘adaptation’

- L.408: which response? Consider merging this sentence with the previous one.

- L.413: pls delete ‘however’

- L.426: pls delete ‘a’ before ‘highlighted’

- L.434: full stop after 2019)

- L.435-436: I did not understand this point. This is unlikely as water deficit induces reductions in cell (vessel) size.

- L.446-447: what general finding?

- L.507: ‘a’ before ‘contributing factor’

First round of review

Manuscript ID: QPB-2024-0070

The paper QPB-2024-0070 presents the effects of repeated periodic droughts on xylem features (vessel and fibre sizes, vessel density, fibre wall thickness) and cambial activity in the supposedly drought-adapted Eucalyptus cladocalyx. Experiments were performed using less than one-year old seedlings subjected to six repeated cycles of water deficit. Measurements relied on xylem histology and microscopic evaluations of stem cross-sections. The authors report highly variable complex patterns depending on the period considered, most likely because of differences in stress intensity and intrinsic genetics.

I believe the paper fits the journal scope and is of potential interest to the reader. The paper clearly has merit by including data from 240 individual seedlings obtained after histological preparation and image analysis for several xylem cell features. However, I see a couple of issues that, so far, prevent the paper from acceptance in my opinion.

1. I did not see why the authors chose to apply such a complex experimental design with six repeated drought cycles rather than a controlled long-lasting moderate drought. Unless I am wrong, I did not see a clear rationale arising from the literature or from hypotheses. I believe such numerous, short and repeated drought events make the histological characterization highly complex so that to identify the exact periods/xylem during which cells were produced (see comment below). It also questions the legacy effects of past drought cycles on xylem cell features as the experiment advances, and this is confounded with cambial seasonal aging.

2. In line with the comment above. Unless I am wrong, I did not see any explanation regarding how measurements of xylem cell features were performed on the trees sampled every three days for examining temporal variations. I mean, what zone between the cambium and the pith was used to make morphometric measurements? Without specific marking or high-resolution growth data, how was this realized to be sure of what was measured was absolutely specific of that period of the experiment with growth constantly varying in response to variable watering treatments? Fig. 9 and 10 kind of show different zones in the xylem corresponding to supposedly different drought cycles, but I could not understand how this was performed either based on the text manuscript or even from the figure legends.

3. Although the authors measured predawn leaf water potential, the implementation of drought in the experiment seems empirical as cycle duration was variable and “adjusted based on visible signs of stress” (see L.119). Which signs? Stress intensity thus also varied from one cycle to the other. The authors actually acknowledge in the conclusion section that differences in stress intensity and duration may explain the variable patterns observed, but this remains speculative.

4. The authors used seedlings which means that each individual was genetically different. Although seeds were obtained from a single stand, genetic diversity remains unknown and I fear that the variable temporal trends obtained from four different seedlings sampled every third day may be partly explained by this issue. In their defence, the authors also acknowledge this issue in the conclusion section (L. 505) but this still raises the question of whether or not the absence of typical patterns for xylem cell features is something “normal” or is masked by the genetic diversity. In addition, from what I understand, morphometric measurements were performed on one single subsample per individual (see L.156, 1/8 of the circular surface). Typically, such measurements are repeated at least two or three times on opposite or distributed radial sectors to account for spatial heterogeneity which can be significant, especially on young seedlings with small stems. This may also explain variation in data.

For these main reasons, I believe the paper cannot be accepted in its present shape. However, the authors may consider a resubmission if they are able to address these issues.

Specifics:

- I found in several instances sentences were too vague. See for instance L.40-42, 59-60 or 76 in the introduction. See also L.396-399 or 406-409 in the discussion.

- L.37: ‘Eucalypts rely heavily on xylem tissue to respond to drought stress’. I believe this can apply to any species. I would remove it.

- L.38-39: ‘as susceptibility to embolism increases’. Pls change by ‘as embolism increases’.

- L.40-43: this deserves to be explained a bit more. What kind of changes?

- L.47: ‘non-uniform xylem developmental patterns’: what is meant here?

- L.52: has instead of have

- L.59-60: what is meant here? Pls specify.

- L.76: better than what?

- L.89-90: hypothesis 3a. I believe this cannot be assessed from the experimental design used in this paper. Potential legacy effects of past drought events accumulate with time such that the effects of multiple repeated droughts and season are confounded.

- L.102-103: the difference in Soil VWC is actually quite important here (1-10%), especially if we consider a sandy soil. Was this meant for control plants before the experiment started? In addition, volumetric soil water content has no real significance per se if the values corresponding to field capacity and permanent wilting are ignored. As such, the 1 to 10% difference could be dramatic in terms of water availability.

- L.138-141: pls indicate the measurements were performed before sunrise.

- L.185: ‘Software and image processing’: I think this part would deserve additional information. Pls also see one of my previous comment regarding the methodology for determining the xylem areas related to each specific drought period.

- L.219: ‘cambial’: cambium? Cambial zone?

- L.269: ‘Overall’: not necessary

- L.283-289: I found this part hard to follow, probably because there is no reference to any figure and the patterns seem very variable. Wonder whether this is necessary.

- Table 3: pls adjust the number of digits for the error related to diameter and height for the PI seedlings.

- L.298 and 326: ‘Radial variations’. Not sure this is actually radial variations that are presented. Once again, from what I understand, these are the temporal trends estimated from punctual individual measurements during the whole experiment, but measurements were systematically made from the same xylem area right below the cambial zone?

- L.320-325: I understand that vessel area might be more precise than vessel diameter, especially if vessels are elliptical. However, I think additional information about corresponding vessel average diameter might be useful just for the sake of comparison and simplicity.

- L.347-348: here and elsewhere, careful, only present significant patterns.

- L.364-365: same comment, avoid ‘weakly significantly’ when the p-value is higher than the cut-off.

- L.396-399: too vague, pls reword.

- L.401 ‘which was the case in our observations’: you cannot write this directly since you did not compare your findings with larger trees.

- L.407-409: too vague, pls reword.

- L.504: becomes

- Fig.2: pls add ‘predawn’ before water potential

- Fig.3: I am not sure this figure is needed since it is somehow replicated in Figs 5, 6, 7 and 8. Also, pls convert kPa in MPa in all figures.

- Fig.4: same comment as for Fig.3. If this one is conserved, it might be worth adding in the back some colour to represent start and end days for all drought cycles (as actually made in Figs 5, 6, 7 and 8).

- Figs 5, 6, 7 and 8: from what I understand, at each date (i.e. every third day), two seedlings per block and per treatment were sampled which makes four seedlings per treatment if I am right. What do individual points in b) and c) panels refer to then? Also, why is the number of points in b) and c) panels for vessel density (Fig.6) less numerous than for other traits (Fig. 5, 7 and 8)?

- Figs 9 and 10: I did not understand how the different periods were delimited (pls see my previous comments). I also do not understand what the three central images represent. What are the arrows for?

---

## [Editor Report]

Dear Dr. Drew,

Your revised manuscript has been reviewed by the two reviewers who already commented on the original version. One of them appreciated that this new version is much improved, but still has minor issues. Please take them into account. Of course, you can not change anything to the intrinsic limits of your experimental design. Any experiment has limits, and despite of them your results are of great value for the field.

In addition to the reviewer’s comments, I propose a few minor corrections:

* L255: Should be Equation 4. Please give a name to the “the rate of enlargement or wall thickening” (for instance rG), such that Equation becomes a real equation.

* The abbreviation “CSA” appears lines 317, 319 and 324, while its signification (cross-sectional area) does not appear before line 424. Please expand the abbreviation on first occurrence.

* L331: “the vessel frequency of wood was 137.87mm²” This does not look like a unit of frequency.

I look forward to receiving your revised manuscript.

---

## [Editor Report]

Thank you for this revised version of the manuscript, which addresses all issues raised by the reviewer. I am glad to endorse its publication.